# Cell-Free Methylated *PTGER4* and *SHOX2* Plasma DNA as a Biomarker for Therapy Monitoring and Prognosis in Advanced Stage NSCLC Patients

**DOI:** 10.3390/diagnostics13132131

**Published:** 2023-06-21

**Authors:** Michael Fleischhacker, Erkan Arslan, Dana Reinicke, Stefan Eisenmann, Gerit Theil, Jens Kollmeier, Christoph Schäper, Christian Grah, Frank Klawonn, Stefan Holdenrieder, Bernd Schmidt

**Affiliations:** 1Klinik für Innere Medizin—Schwerpunkt Pneumologie und Schlafmedizin, DRK Kliniken Berlin/Mitte, 13359 Berlin, Germany; b.schmidt@drk-kliniken-berlin.de; 2Lungenarztpraxis Berlin-Reinickendorf, 13403 Berlin, Germany; praxis@erkan-arslan.de; 3Department für Innere Medizin, Universitätsklinikum Halle/Saale, 06120 Halle (Saale), Germany; dana.reinicke@ukh-halle.de (D.R.); stefan.eisenmann@ukh-halle.de (S.E.); gerit.theil@ukh-halle.de (G.T.); 4Lungenklinik Heckeshorn, Helios Klinikum Emil von Behring, 14165 Berlin, Germany; jens.kollmeier@helios-gesundheit.de; 5Klinik und Poliklinik für Innere Medizin B, Universitätsmedizin Greifswald, 17475 Greifswald, Germany; christoph.schaeper@jsd.de; 6Gemeinschaftskrankenhaus Havelhöhe, Pneumologie und Lungenkrebszentrum, 14089 Berlin, Germany; christian.grah@havelhoehe.de; 7Department of Computer Science, Ostfalia University, 38302 Wolfenbüttel, Germany; klawonn@helmholtz-hzi.de; 8Biostatistics, Helmholtz Centre for Infection Research, 38124 Braunschweig, Germany; 9Munich Biomarker Research Center, Institute of Laboratory Medicine, German Heart Centre, Technical University Munich, Lazarettstraße 36, 80636 Munich, Germany; s.holdenrieder@tum.de

**Keywords:** liquid profiling, plasma, methylation, *mSHOX2*, *mPTGER4*

## Abstract

Notwithstanding some improvement in the earlier detection of patients with lung cancer, most of them still present with a late-stage disease at the time of diagnosis. Next to the most frequently utilized factors affecting the prognosis of lung cancer patients (stage, performance, and age), the recent application of biomarkers obtained by liquid profiling has gained more acceptance. In our study, we aimed to answer these questions: (i) Is the quantification of free-circulating methylated *PTGER4* and *SHOX2* plasma DNA a useful method for therapy monitoring, and is this also possible for patients treated with different therapy regimens? (ii) Is this approach possible when blood-drawing tubes, which allow for a delayed processing of blood samples, are utilized? Baseline values for *mPTGER4* and *mSHOX2* do not allow for clear discrimination between different response groups. In contrast, the combination of the methylation values for both genes shows a clear difference between responders vs. non-responders at the time of re-staging. Furthermore, blood drawing into tubes stabilizing the sample allows researchers more flexibility.

## 1. Introduction

The last years have seen some progress in the treatment of advanced-stage lung cancer patients, specifically, for patients harboring druggable genetic alterations like activating *EGFR* gene mutations, mutations in *BRAF* and *MET* genes, and variants of *ALK* and *ROS1* [1]. This applies to non-small cell (NSCLC) and small-cell (SCLC) lung cancer patients who are treated with targeted or immunotherapeutic agents [2]. Notwithstanding, different resistance mechanisms are in place, leading to a therapy failure for these novel treatments as well. This underscores the need for searching out and establishing reliable tumor markers for therapy monitoring.

Recently, liquid profiling has emerged as a promising tool for a longitudinal analysis of response to a given therapy, particularly because it can be performed in real time. After the first description of cell-free nucleic acids detectable in human plasma and serum [3], the group of Anker and Stroun laid the foundation for the current interest in this method [4,5,6,7]. Currently, several methods exist for the analysis of extracellular nucleic acids quantitatively and qualitatively with whole genome sequencing, exon sequencing, genome-wide methylation analysis, and others [8,9].

An easy, fast, and straightforward approach is the quantitative measurement of a smaller gene panel of methylated sequences in cell-free DNA with real-time PCR. The usefulness of this approach to demonstrate a relationship between the presence of methylated plasma DNA and therapy response in lung cancer patients has been shown in several papers [10,11,12,13].

Together with Wang et al. [14], our group was among the first in demonstrating the potential of a longitudinal analysis of methylated, cell-free DNA for therapy monitoring. We demonstrated a good correlation between the longitudinal measurement of extracellular plasma *mSHOX2* DNA and the response to cytotoxic treatment in late-stage lung cancer patients [15]. In this study we used EDTA tubes processed within one to two hours after blood drawing. Previous studies had demonstrated that EDTA blood should not be stored for more than six hours because longer storage leads to cell lysis and a ”contamination“ of cell-free DNA with genomic DNA. To allow for an extended storage of blood tubes (including shipping to a remote laboratory), it is necessary to stabilize the blood sample with special additives. Several companies have developed specially designed blood-drawing tubes such as the PAXgene Blood ccfDNA tubes that received market approval in 2016.

In this investigation, we extended our former study as we included more patients from several hospitals (multicenter) and utilized PAXgene Blood ccfDNA tubes (Qiagen, Hilden, Germany) for blood sampling. Furthermore, we employed a modified marker panel for the detection of methylated cell-free circulating DNA in the plasma of late-stage lung cancer patients undergoing systemic therapy.

## 2. Materials and Methods

### 2.1. Patients

In total, 96 patients, among them 78 from the DRK Kliniken Berlin-Mitte and 18 from the other participating clinics, with late-stage histologically confirmed non-small cell lung cancer (NSCLC) who received a first-line treatment consisting of chemo +/− radiotherapy, anti-*EGFR* therapy, or immunotherapy were enrolled prospectively in the present study. The patients were consecutively referred to the participating clinics for diagnosis and treatment from August 2016 to October 2020. The study was approved by the Ethics Committee of the University Halle/Saale on 19 June 2014 (Reference number 2014-52), and all patients gave informed written consent prior to inclusion in the study. This study was set up as a multicentric trial, and most patients were treated at the DRK Kliniken Berlin-Mitte (78 patients). The blood of these 78 patients was drawn in PAXgene Blood ccfDNA tubes (Qiagen, Hilden, Germany), while EDTA tubes were used for the 18 patients treated in the other participating centers. For the evaluation of the therapy response, the data from all 96 patients were used, while the analysis for the prognostic relevance was limited to the 78 patients enrolled at the DRK Kliniken Berlin-Mitte.

The details of the clinical data of the patients are shown in Table 1. The re-staging was done after the first two therapy cycles of chemo +/− radiotherapy or 6–8 weeks after start of therapy, respectively. The response evaluation was carried out according to the RECIST v1.1 criteria.

### 2.2. Plasma Preparation

Blood was taken from the patients at the time of diagnosis (pre-treatment) and thereafter in intervals of 7 to 10 days until the time of re-staging. The samples obtained at the DRK Kliniken Berlin-Mitte were collected in PAXgene Blood ccfDNA tubes and shipped by regular mail or transported via courier (at room temperature in both instances) to the laboratory in Halle/Saale. The range of delay in processing the blood was 0 to 8 days (median 5 days). The blood was spun once at 1500 rpm for 10 min, and the supernatant was carefully transferred into a new tube and re-centrifuged at 3500 rpm for 10 min. The cell-free plasma was aliquoted and stored at −80 °C before processing. The blood samples drawn at the other three participating study centers were processed as described above, and the plasma was stored at −80 °C and shipped on dry ice to the laboratory in Halle/Saale.

### 2.3. DNA Isolation, Purification, and Bisulfite Conversion

We employed the Epi Bis-kits (Epigenomics, Berlin, Germany) according to the protocol of the manufacturer with an initial input of 3.5 mL plasma for the isolation, purification, and bisulfite treatment of the DNA. For the detection and quantification of the gene products, a PCR kit developed by Epigenomics for the simultaneous quantification of *mSHOX2*, *mPTGER4,* and *ß-actin* as a reference gene [16] was employed. All samples were measured in triplicate. The quantification of *mSHOX2, mPTGER4* and *ß-actin* (as the reference gene for the calculation) was performed according to Kneip et al. [17]. The quantification of the cell-free methylated sequences was performed after all prospectively collected plasma samples were complete, i.e., making this analysis an observational study.

### 2.4. Statistical Analysis

The distribution of *PTGER4* and *SHOX2* methylation values is demonstrated as boxplots for all response groups at the time of first radiological staging; the values were partial remission (PR), stable disease (SD), and progressive disease (PD). The prediction of the therapy response was evaluated in two settings: For detection of progression, PD was compared with SD + PR, while for detection of response, PR was compared with SD + PD. The power of discrimination was calculated as an area under the curve (AUC) of the receiver-operating characteristic (ROC) curves, and sensitivity was recorded at 90% specificity. The significance of statistical differences was calculated by the Wilcoxon-test. For the combination of markers, decision-tree models were established utilizing appropriate cutoffs and marker order, and probabilities for therapy response were shown at every node. Survival was evaluated by the Kaplan-Meier-curves and log-rank test. All comparisons were performed two-sided, and statistical significance was set at *p* < 0.05. Data analysis was performed using R (version 4.2.0; https://www.R-project.org, (accessed on 15 February 2023), free software foundation, Inc., Strongsville, OH, USA).

## 3. Results

All patients received serial sample collections during the treatment, with an average of 6 samples per patient. Treatment was performed for a minimum of 6 months; the median follow-up of the patients was 22 months, with a range from 6 to 58 months. We evaluated the biomarker values before the start of therapy and at re-staging exams; the relative changes were considered and correlated with the radiological response to treatment and the survival of the patients.

Of those 96 patients with advanced NSCLC, 83 received regular chemo +/− radiotherapy, 9 received a combination of chemo- and immunotherapy, and 4 patients, immuntherapy only. Thirty-four patients responded well to the treatment at staging exams and showed partial remission (PR), 15 had stable disease (SD), and 33 had progressive disease (PD). The therapy response of 14 patients is unknown. The median overall survival (OS) was 8.5 months with a range from 1 to 37 months (Table 1).

### 3.1. Distribution of mPTGER4 and mSHOX2 Methylation Values in Response Groups

Before the start of therapy (V1), patients who showed partial remission had slightly lower methylated *PTGER4* levels than SD and PD patients, but this difference was not statistically significant. However, at the time of the re-staging exam (VS), *mPTGER4* levels of PR patients were significantly lower than those of SD and even more than those of PD patients—and the ratio of *mPTGER4* VS/V1 was also significantly different among the response groups (Figure 1A–C). Similar tendencies were observed for *SHOX2* methylation with no significant differences between response groups for the pre-therapeutic assessment and lower values for PR and SD than for PD patients. In contrast, comparable to the *mPTGER4* values, the differences seen for the ratio of *mSHOX2* at VS and VS/V1 values reached a low significance level (Figure 1D–F).

### 3.2. Differentiation of Patients According to Their Therapy Response

The power of discrimination between patients with good and poor response to therapy was objectified by areas under the curves (AUC) of receiver operating characteristic (ROC) curves.

For the comparison of patients with progression (PD) versus no progression (PR + SD), AUCs for *mPTGER4* were AUC = 0.60 (V1), AUC = 0.72 (VS), and AUC = 0.66 (VS/V1) and for *mSHOX2* AUC = 0.57 (V1), AUC = 0.67 (VS), and AUC = 0.71 (VS/V1). Sensitivities for the detection of progression at a 90% specificity vs. non-progression were for *mPTGER4* 0.60 at V1, 0.74 at VS and 0.65 for VS/V1 (Figure 2A–C) as well as for *mSHOX2* 0,59 at V1, 0.64 at VS and 0.69 for VS/V1, respectively (Figure 2D–F). For the comparison of patients with remission (PR) at a 90% specificity versus no remission (SD + PD), AUCs for *mPTGER4* were AUC = 0.60 (V1), AUC = 0.74 (VS), and AUC = 0.65 (VS/V1) (Figure 3A–C) and for *mSHOX2* AUC = 0.59 (V1), AUC = 0.64 (VS), and AUC = 0.69 (VS/V1) (Figure 3D–F).

We also tested whether the patient’s age and sex had any influence on the therapy response but did not find any correlation.

### 3.3. Combination of Markers in Decision Trees

To maximize the information for a therapy prediction, markers were combined in a decision-tree model by employing appropriate cutoffs and marker order. In using this model, we demonstrated that patients with a low *mSHOX2* ratio (VS/V1) and low *mPTGER4* levels at the time of staging (VS) had a high chance of achieving PR at staging exams (Figure 4, Node 3). In contrast, patients with a high *mSHOX2* ratio (VS/V1) and a high *mPTGER4* ratio (VS/V1) had a high probability of demonstrating a progressive disease (Figure 4, Node 7). Patients with either a low *mSHOX2* ratio (VS/V1) and high *mPTGER4* (VS) levels or, alternatively, a high *mSHOX2* ratio (VS/V1) and a low *mPTGER4* ratio (VS/V1) were in the intermediate range(Figure 4, Nodes 4 and 6).

### 3.4. Relevance of PTGER4 and SHOX2 for Prognosis

Information regarding overall survival (OS) was available for 78 out of the 96 NSCLC patients. For prediction of OS only, *SHOX2* methylation at the time of staging (VS) showed borderline-significant prognostic information. All other markers and time points did not. Patients with *SHOX2* methylation levels below the median tended to longer survival with a median OS of 11 months as compared with patients with *SHOX2* methylation levels above the median who had a median OS of 8 months. This difference showed a trend but was not statistically significant with a value of *p* = 0.058 (Figure 5).

## 4. Discussion

According to the latest US cancer statistics, 46% of the newly diagnosed lung cancer patients demonstrate a late-stage disease [18]. Furthermore, a recent survey showed that almost 49% of 210,000 newly diagnosed lung cancer patients died within 2 months after receiving their diagnosis [19]. Thus, it is of utmost importance to differentiate between patients with a high risk for early death and patients with a better prognosis.

Since our group and other researchers demonstrated that the detection and quantification of methylated ctDNA is a useful approach for therapy monitoring in lung cancer patients [14,15,16], more papers on this subject were published [20,21]. Moreover, the incorporation of liquid profiling as an additional clinical tool for the diagnosis, treatment stratification, detection of resistance mechanisms, and prognostic indication in lung cancer patients has been shown [22,23,24,25].

To confirm and extend the results of our original analysis [15] and to demonstrate the robustness of the method, we performed the present study. Apart from increasing the number of patients, we included patients who had received different treatment regimens, i.e., chemotherapy with/without radiotherapy, chemotherapy plus immunotherapy, and immunotherapy exclusively. Furthermore, we changed the pre-analytical process and applied a modified marker panel.

It is known that blood drawn into EDTA tubes should not be stored for more than 4–6 h before being processed [26]. A delay of more than 6 h can lead to irregular results due to lysis of blood cells and “contamination” of cell-free DNA with genomic DNA. Several studies have demonstrated that storage of blood drawn into PAXgene Blood ccfDNA Tubes for up to 7 days does not change the quantity and quality of plasma DNA [27,28].Moreover, we applied a modified marker panel which included the detection of cell-free methylated *SHOX2* and *PTGER4* plasma DNA [27]. All these factors plus a different patient population might explain the different results obtained in this study compared to those of our previous paper [15].

It is interesting to note that the baseline values for *mPTGER4* and *mSHOX2* do not allow for a clear discrimination between different response groups (Figure 1). In contrast, the methylation values for both genes show a clear difference between responders vs. non-responders at the time of re-staging (Figure 1). This observation still holds true when the ratios of the methylation values (VS/V1) for both genes are plotted (Figure 1E,F). These data corroborate our initial observation [15] that the methylation values at the time of diagnosis do not allow a differentiation between the two groups, while this is possible at the time of re-staging (i.e., 8 to 12 weeks after therapy start). When this study was commenced, Epigenomics AG (Berlin, Germany) had introduced a modified kit for DNA methylation analysis. This new kit included *mSHOX2* as well as a second marker, *mPTGER4* [27]. Currently, no comparative data exist for the two kits in a head-to-head approach. However, promising results have been published that advocate for the inclusion of a second marker. Indeed, the grouping of patients with a low *mSHOX2* ratio (VS/V1) plus low *mPTGER4* levels at the time of re-staging (VS) in decision trees allowed the discrimination of patients responding to the therapy from non-responding patients (Figure 3). When patients with a high *mSHOX2* ratio (VS/V1) and high *mPTGER4* levels at the time of staging (VS) were combined, we were able to differentiate patients not responding to the therapy, but with less statistical power. When the methylation levels of both genes at the time of re-staging were used for Kaplan-Meier curves only, *mSHOX2* demonstrated a trend for statistical significance, but not *mPTGER4* (Figure 4). *SHOX2* belongs to the homeobox family and is coding for a transcriptional regulator involved in pattern formation in both the invertebrate and vertebrate species. In contrast, *PTGER4* is a protein-coding gene and belongs to the G-protein coupled receptor family. As for both genes, no role in the development of lung cancer has been described so far; thus, we assume that only the former has a functional relationship in lung cancer patients.

In conclusion, our findings demonstrate that quantifying extracellular free-circulating methylated DNA in plasma could be a valuable tool to monitor the response of lung cancer patients undergoing various treatment regimens. The use of specialized blood- drawing tubes to stabilize samples at room temperature allows their collection and transport to a remote laboratory without cooling. This approach gives researchers the possibility of enrolling many patients in future studies to confirm the validity of our current findings.

## Figures and Tables

**Figure 1 diagnostics-13-02131-f001:**
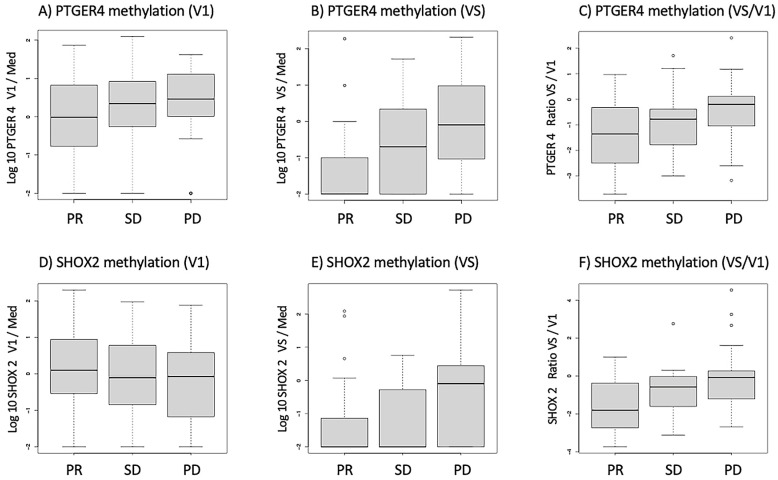
Boxplots for the distribution of methylated *PTGER4* (**A**–**C**) and *SHOX2* (**D**–**F**) before the start of therapy (V1), at the time of first radiological staging (VS) and the ratio between both time points (VS/V1) for patients with partial remission (PR), stable disease (SD), and progressive disease (PD) in the staging exams.

**Figure 2 diagnostics-13-02131-f002:**
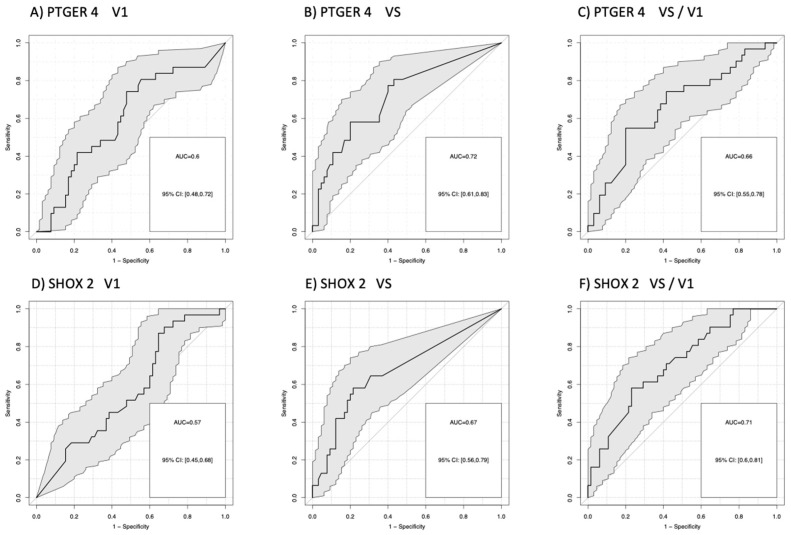
Receiver-operating characteristic (ROC) curves (dark line) and respective 95% confidence intervals (CI; grey area) for the discrimination of patients with progressive disease (PD) from patients with no progression (stable disease + partial remission) for methylated *PTGER4* (**A**–**C**) and *SHOX2* (**D**–**F**) before start of therapy (V1), at time of first radiological staging (VS) and the ratio between both time points (VS/V1).

**Figure 3 diagnostics-13-02131-f003:**
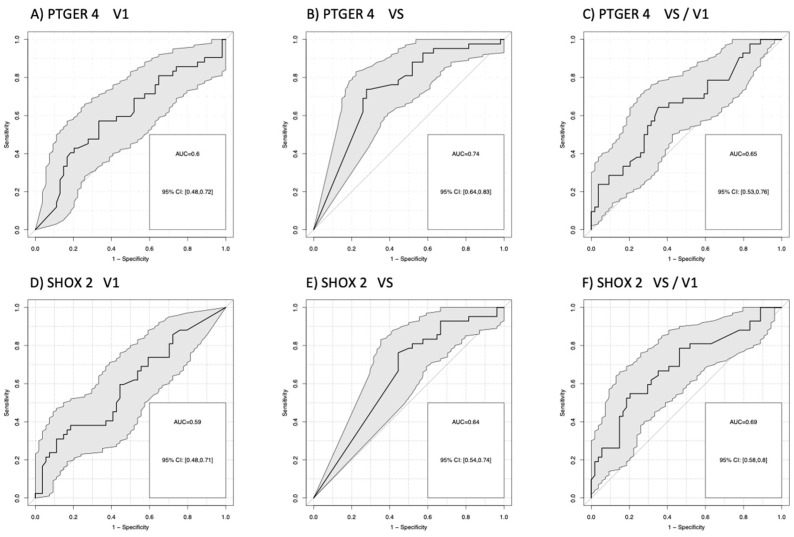
Receiver-operating characteristic (ROC) curves for the discrimination of patients with remission (PR) from patients with no remission (stable + progressive disease) for methylated *PTGER4* (**A**–**C**) and *SHOX2* (**D**–**F**) before start of therapy (V1), at time of first radiological staging (VS) and the ratio between both time points (VS/V1).

**Figure 4 diagnostics-13-02131-f004:**
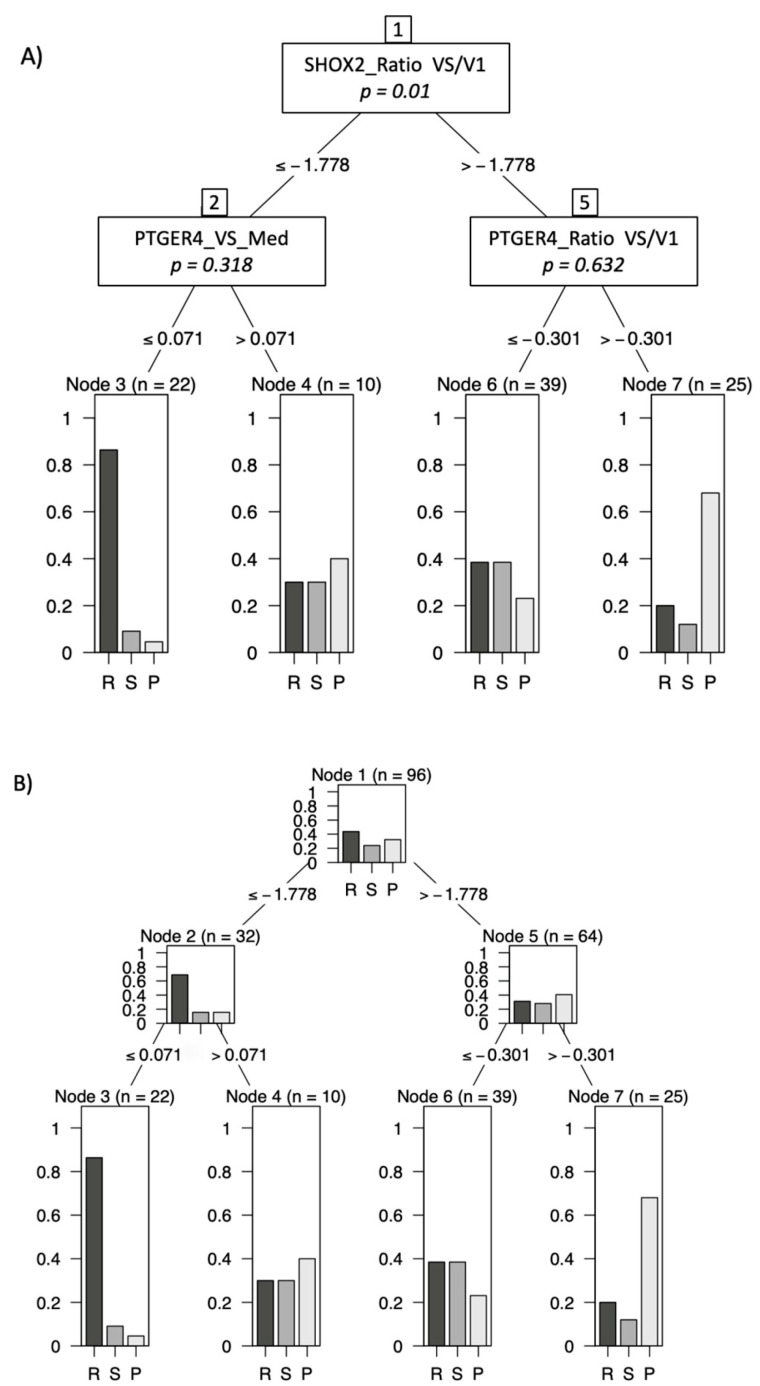
Decision-tree model using a combination of markers *mSHOX2* ratio (VS/V1), *mPTGER4* at staging (VS) and *mPTGER4* ratio (VS/V1) for best prediction of therapy response. Patients with low *mSHOX2* VS/V1 and low *mPTGER4* VS had a high chance of achieving partial remission (R), whereas patients with high *mSHOX2* VS/V1 and high *mPTGER4* VS/V1 had a high probability for progressive disease (P). (**A**) shows the decision rules; and (**B**) the distribution of responses at every decision node.

**Figure 5 diagnostics-13-02131-f005:**
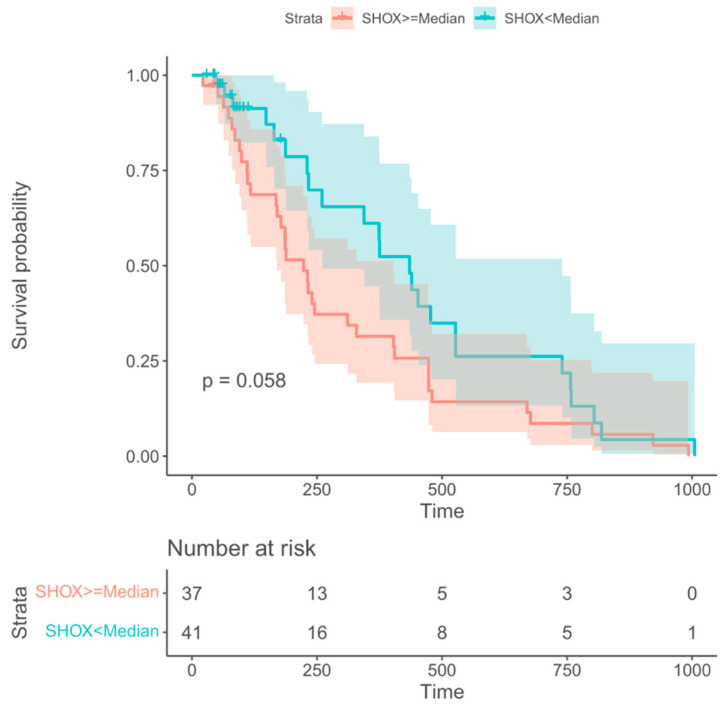
Kaplan-Meier survival curves for the *mSHOX2* at time of staging (VS) showing borderline longer survival for patients with low levels (green) than those with high levels (red).

**Table 1 diagnostics-13-02131-t001:** Clinical data for all enrolled patients.

Patient	Number
Sex	
female	62
male	34
Age	
range	48–81
median	63
Smoker	
yes	41
no	4
ex-smoker	30
unknown	21
Histology	
NSCLC (unclassified)	25
Adenocarcinoma	54
Squamous cell carcinoma	17
Treatment	
Chemo +/− Radiotherapy	83
Chemo + Immunotherapy	9
Immunotherapy	4
Therapy response	
partial remission	42
stable disease	23
progressive disease	31

## Data Availability

Not applicable.

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
