# Peer review of "Cell-Free Methylated *PTGER4* and *SHOX2* Plasma DNA as a Biomarker for Therapy Monitoring and Prognosis in Advanced Stage NSCLC Patients"

_diagnostics, 2023, doi:10.3390/diagnostics13132131_

Round 1
Reviewer 1 Report
In this manuscript, Michael et al. collected the sample and investigate the methylated PTGER4 and SHOX2 in non-small cell lung cancer. They demonstrate that free-circulating methylated DNA in plasma could be a valuable tool to monitor the response of NSCLC. They also demonstrate that the blood drawing tubes can be used for a delayed processing of blood samples. The results are good but some improvements are needed.
Comments for the authors:
1. Table 1, please use the standard table format for research manuscript
2. Figure 2, Figure legend for three different lines in each figure
3. Do the authors upload their quantified data to the public website?
4. Please provide more details for draw ROC curve. In my opinion, drawing an ROC curve involves many processes.
5. When I search “PTGER4 and SHOX2 methylation in lung cancer” in PubMed, many papers reported or demonstrated PTGER4/SHOX2 methylation in NSCLC, what is improvement of the authors’ work compared to previous works?
The following two papers are for example of similar works
https://www.ncbi.nlm.nih.gov/pmc/articles/PMC8373977
https://pubmed.ncbi.nlm.nih.gov/33662689/
none
Author Response
Q 1. Table 1, please use the standard table format for research manuscript
Answer
Thank you very much. We have adjusted the table according the journal style.
Q 2. Figure 2, Figure legend for three different lines in each figure
Answer
We have specified the legend as follows:
Receiver-operating characteristic (ROC) curves (dark line) and respective 95% confidence intervals (CI; grey area) for the discrimination of patients with 187 progressive disease (PD) from patients with no progression (stable disease + partial remission) for 188 methylated PTGER4 (A-C) and SHOX2 (D-F) before start of therapy (V1), at time of first radiological 189 staging (VS) and the ratio between both time points (VS/V1).
Q 3. Do the authors upload their quantified data to the public website?
Answer
The data is not uploaded to a public website but can be made available upon specific request.
Q 4. Please provide more details for draw ROC curve. In my opinion, drawing an ROC curve involves many processes.
Answer
An receiver operating characteristic (ROC) curve plots the whole profile of sensitivity versus 1-specificity between two groups of patients over all possible cutoffs of the parameter of interest. In our paper it shows the discriminatory power between the groups of patients with progression vs patients with no progression during therapy by use of the methylation markers. In addition, the 95% confidence interval (CI) is calculated to show the certainity of the information. If CI does not include 0.5, then the discrimination is significant with a p<0.05. Software to calculate ROC curves it is easily available on https://www.r-project.org or https://cran.rstudio.com and there are also multiple descriptions how to do it.
Q 5. When I search “PTGER4 and SHOX2 methylation in lung cancer” in PubMed, many papers reported or demonstrated PTGER4/SHOX2 methylation in NSCLC, what is improvement of the authors’ work compared to previous works?
The following two papers are for example of similar works
https://www.ncbi.nlm.nih.gov/pmc/articles/PMC8373977
https://pubmed.ncbi.nlm.nih.gov/33662689/
Answer
All the papers showing up using this search strategy use the methylation of PTGER4 and/or SHOX2 for the discrimination of lung cancer patients from healthy or benign control groups, i.e. are of diagnostic relevance of methylation for lung cancer patients.
However, our study question was related to response monitoring and prognosis in NSCLC patients which is a completely different approach. Here we investigated whether patients with tumor progression during systemic therapy can be discriminated from non-progressing tumor patients. This would possibly allow a more certain or earlier change of the treatment. In addition we investigated the survival probability (prognosis) if patients are stratified by these methylation markers. Regarding these clinical questions, there are currently no publications available.
Reviewer 2 Report
The authors sort to confirm and extend the results of their previous study here by determining whether (i) quantification of free-circulating methylated PTGER4 and SHOX2 plasma DNA is useful for monitoring patients receiving different therapy regimens, and (ii) if the approach is feasible if sample processing is delayed, albeit provided the use of PAXgene Blood ccfDNA Tubes. This study uses a larger cohort to confirm that these markers may be useful in distinguishing responders vs non-responders during re-staging. Overall, the study provides value to this growing field, albeit further studies using larger cohorts are needed. Few minor comments:
-Did the authors compare the markers between the treatment groups? Were there any treatment-induced differences?
-Were there any differences between gender, patient or between smokers vs non-smokers?
Author Response
- -Did the authors compare the markers between the treatment groups? Were there any treatment-induced differences?
Answer
In the present manuscript we performed a comparison of patients with advanced NSCLC who had a progression of tumor disease during systemic treatment and patients who did not suffer from progressive tumor objectified by radiological staging. Most patients received chemo +/- radiotherapy. As there was only a small group of patients receiving immunotherapy this could not be evaluated separately.
Q -Were there any differences between gender, patient or between smokers vs non-smokers?
Answer
In the present analysis, we only tested the power of the biomarkers for the discrimination of response to treatment. Interaction of age and gender was not observed. Number of non-smokers was too small for separate testing.
This sentence was added to the MS: We also tested whether patient's age and sex had any any influence on the therapy response but did not find any correlation. (line 186- 187)
Round 2
Reviewer 1 Report
Accept